# An In Vitro Evaluation of Denture Cleansing Regimens against a Polymicrobial Denture Biofilm Model

**DOI:** 10.3390/antibiotics11010113

**Published:** 2022-01-16

**Authors:** Jason L. Brown, Tracy Young, Emily McKloud, Mark C. Butcher, David Bradshaw, Jonathan R. Pratten, Gordon Ramage

**Affiliations:** 1Oral Science Research Group, Glasgow Dental Hospital, School of Medicine, Dentistry and Nursing, College of Medical, Veterinary and Life Sciences, Glasgow University, Glasgow G2 3JZ, UK; t.young.1@research.gla.ac.uk (T.Y.); e.mckloud.1@research.gla.ac.uk (E.M.); m.butcher.1@research.gla.ac.uk (M.C.B.); gordon.ramage@glasgow.ac.uk (G.R.); 2Oral Healthcare R&D, GlaxoSmithKline Consumer Healthcare, Weybridge KT13 0DE, UK; david.j.bradshaw@gsk.com (D.B.); Jonathan.r.pratten@gsk.com (J.R.P.)

**Keywords:** biofilm model, polymicrobial, denture cleanser, antimicrobials

## Abstract

Denture stomatitis (DS) is an inflammatory disease resulting from a polymicrobial biofilm perturbation at the denture surface–palatal mucosa interface. Recommendations made by dental health care professionals often lack clarity for appropriate denture cleaning. This study investigated the efficacy of brushing with off-the-shelf denture cleanser (DC) tablets (Poligrip^®^) vs. two toothpastes (Colgate^®^ and Crest^®^) in alleviating the viable microorganisms (bacteria and fungi) in an in vitro denture biofilm model. Biofilms were grown on poly(methyl)methacrylate (PMMA) discs, then treated daily for 7 days with mechanical disruption (brushing), plus Poligrip^®^ DC, Colgate^®^ or Crest^®^ toothpastes. Weekly treatment with Poligrip^®^ DC on day 7 only was compared to daily modalities. All treatment parameters were processed to determine viable colony forming units for bacteria and fungi using the Miles and Misra technique, and imaged by confocal laser scanning microscopy (CLSM). Brushing with daily DC therapy was the most effective treatment in reducing the viable biofilm over 7 days of treatment. Brushing only was ineffective in controlling the viable bioburden, which was confirmed by CLSM imaging. This data indicates that regular cleansing of PMMA with DC was best for polymicrobial biofilms.

## 1. Introduction

Denture stomatitis (DS) is an oral disease affecting up to 70% of older patients [1]. DS is the inflammation caused by contact of the denture to the oral mucosa and is highly associated with the cleanliness of the denture among other risk factors [2,3]. *C. albicans* has been identified as the main etiological agent in the progression of DS [4]. It has been detected on the surface of dentures in 72% of patients, though often with no sign of DS [5,6]. This indicates that while *Candida* species prevalence may be a risk factor, the bacterial microbiome of the denture may also influence disease. Indeed, conventionally, DS is treated with fluconazole, itraconazole, or nystatin (amphotericin B). These antifungal drugs have a limited scope against *Candida* biofilms, and no activity against the dominant bacterial species within these communities [7,8]. Moreover, there is evidence that interkingdom interactions play a critical role in protecting fungal and bacterial species from antibiotics and antifungals [9,10]. Recent research has highlighted the complexity of the biofilms present on the surface of dentures [6,11,12,13]. Thus, in vitro biofilm models investigating DS need to reflect the interkingdom dynamic for accurate and representative antimicrobial testing of oral healthcare products, such as denture cleansers and toothpastes. Such a consideration is often overlooked for these model testing platforms.

Consumer studies indicate a wide variety of techniques used to clean dentures with varied success [14,15], including mechanical brushing, effervescent tablets, toothpastes, bleach, soap and microwave disinfection. Clear and concise recommendations as to the best practice in denture cleaning is important, yet often overlooked [16]. Therefore, studies utilising guidelines based upon clinical studies are paramount. For example, our group integrated an in vitro testing regimen to inform a clinical trial design. We demonstrated the need for daily versus intermittent denture cleaning. Intermittent, when compared to daily treatment, was less effective, both in vitro and in vivo, based on clinical and microbiological parameters [17]. Other solely in vitro studies have investigated the efficacy of denture cleansing of relevant multi-species biofilms, with varying results [18,19,20]. Such studies have either been limited in the microbial consortia used, applied different methodologies to the study design, or failed to compare the efficacy of treatment modalities. Of note, a previous publication by our research group corroborated the observation from above that daily cleansing of the denture biofilm was more effective than weekly cleansing, although such work was limited to the one antimicrobial treatment [20].

Here, the main purpose of this study was to evaluate the effectiveness of mechanical disruption and a denture cleanser (Poligrip^®^) vs. two standard fluoride-containing toothpaste treatments, on a bespoke polymicrobial denture biofilm model. Treatment efficacy was determined using viable cell counting and confocal imaging. The null hypothesis of the study was that no difference would be detected in the efficacy of the various treatment modalities tested against the polymicrobial biofilm model.

## 2. Results

### 2.1. Combined Brushing with Daily Denture Cleansing Reduces Biofilm Bioburden

Colony forming unit (CFU) counts for aerobic (Figure 1A), anaerobic bacteria (Figure 1B) and fungi (Figure 1C) were determined by plate counting over the 7-day treatment in the non-treated controls on all days the CFU counts were above ~5 × 10^7^. Anaerobic bacteria were significantly reduced from day 1 to day 7 (** *p* < 0.01 or *** *p* < 0.001). Both toothpaste treatments were mildly effective in ameliorating the anaerobic cell counts, and mostly comparable in efficacy. Brushing was the least efficient method for biofilm removal. CFU counts from 1 weekly DC treatment (denoted by the dotted line) did not significantly differ from non-treated controls.

Similar trends in efficacy were observed for the treatment modalities against aerobic bacteria and fungi. Brushing and DC therapy was the most potent, reducing the viable bacterial load of the biofilm from ~5 × 10^7^ CFU/mL in non-treated controls by between 2 and 5 logs across the entire experiment (** *p* < 0.01 or *** *p* < 0.001, respectively). Aerobic CFUs were significantly reduced intermittently by the Colgate^®^ and Crest^®^ treatment over the treatment period. Brushing only was unsuccessful in decreasing the viable cell counts across all days, as was weekly DC treatment on day 7, when compared to non-treated controls.

*C. albicans* colonization of the biofilms ranged from ~1 × 10^7^ CFU/mL to 1 × 10^9^ CFU/mL in non-treated controls. As with the bacteria, viable fungal counts were significantly reduced most by DC + brushing dual therapy, although this did not reach significance on day 2 and day 5. The two toothpastes and brushing only were largely ineffectual against the biofilms across all days, except for Colgate^®^ treatment on day 4, which significantly decreased the fungal counts (* *p* < 0.05). Weekly DC treatment did not reduce fungal CFUs on day 7, with daily DC usage significantly better at controlling the biofilm bioburden.

### 2.2. Image Analysis Demonstrates Superior Physical Benefits of Daily Denture Cleansing

To corroborate the observations from the CFU counts, confocal laser scanning end point images were gathered for non-treated and treated biofilms on day 7 (Figure 2). Control biofilms covered the PMMA surface with heavily populated micro-colonies. As noted by the bright fluorescent green stain (SYTO™ 9). Complete eradication of the biofilm was not observed with any treatment. Conversely, treatment with DC with brushing heavily disrupted the biofilm, leaving few colonies. Brushing only has minimal effect on the biofilm, with images comparable to non-treated biofilms. Brushing with Colgate^®^ and Crest^®^ had an intermittent effect, leaving sparse pockets of viable cells on the PMMA disc, in agreement with the CFU counts. Minimal red propidium iodide (PI) staining was detected, suggestive that all biofilms remaining on the discs were largely viable. DC and brushing therapy were far superior to other treatments in reducing viable bacterial and fungal counts in the denture biofilm model.

## 3. Discussion

The use of dentures occurs with an aging population. Dentures have a large surface area with indentations and pores that can harbour biofilms, where viable organisms remain [11]. Brushing plus chemical irrigation, were successful at reducing the microbial load. This emphasizes the need for regular brushing and treatment of the denture material to minimize bioburden, which can drive inflammatory diseases such as DS.

The complex biofilm utilised in this study challenged the ideology that *C. albicans* is the sole causative agent of DS. Studies have suggested that *Candida* carriage remains a risk factor for DS because the fungal load can be as high as 8 log_10_ units on patients’ dentures [21]. Our research group has previously shown that DC treatment in vitro can alleviate *C. albicans* biofilm growth on denture material [22]. The presence of a complex polymicrobial biofilm on the denture surface is also an important factor in disease pathogenesis [6,12]. *Candida* is essential for supporting the growth of bacteria. *C. albicans* offers a structural scaffold for other organisms to adhere [10,23]. Therefore, a polymicrobial biofilm model containing both bacteria and fungi was used.

The denture cleanser (Poligrip^®^) used in this study utilises oxidation to produce hydrogen peroxide and active oxygen, which have antimicrobial effects [24]. This oxidative stress targets cell membranes and causes DNA damage and disruption of respiration [25]. During the treatment, all organisms demonstrated levels of resilience despite being reduced compared to non-treated controls, and so may be able to utilise stress pathways in response to treatment. Bacteria and *C. albicans* utilise detoxifying mechanisms such as thioredoxins and superoxide dismutase to reduce these oxidative agents [26,27]. While brushing with DC remained the most successful treatment modality, brushing with toothpastes offered additional benefits over brushing alone, as illustrated by the reduction in large cell aggregates in CLSM. Fluoride was the active agent within both toothpastes (0.15% active ion), however, Colgate^®^ toothpaste contains sodium monofluorophosphate while Crest^®^ contains sodium fluoride as the stabiliser. Conflicting evidence exists as to the most efficacious of the two, as other compounds such as whitening agents (mica or titanium dioxide) or foaming agents (sodium lauryl sulfate) may contribute to their antimicrobial effects [28,29,30]. Within this study, both toothpastes were equally, but imperfectly, less effective in reducing viable bioburden compared to DC. There were benefits to toothpastes as a cleaning method, though it should be noted that additional abrasion may damage the PMMA and reduce the lifespan of the denture material [31,32]. Toothpaste brushing merits further investigation into the long-term effects in controlling denture biofilm formation.

Overall, with adherence to PMMA demonstrated within 4 h for *Candida* and quicker for initial bacterial attachment [22,33], the denture material will never achieve complete sterilisation. The main aim should therefore be to lower the viable microbial load to reduce the risk to any immunocompromised or older patients [34]. This study demonstrates that regular mechanical and chemical intervention is necessary for maintaining a low level of denture plaque, which is particularly important given that significant reduction in microbial bioburden remains the most important aim in combatting DS. The results here prove that the broad-spectrum antimicrobial nature of DC therapy is essential in controlling the polymicrobial community of the denture material. Although this is an exploratory in vitro study, and implications for in vivo testing are not yet clear, we have demonstrated different efficacies in DC treatment regimens that would be intriguing to investigate in future clinical studies. It is important to note that given the porous nature of the denture material, the CFU counts should be taken as estimates, as sonication is unlikely to remove the entire biomass. This can be confirmed in the CSLM micrographs.

## 4. Conclusions

Denture cleaning treatment impacts polymicrobial interkingdom growth on PMMA denture material. Daily brushing with DC therapy remains the most effective treatment in alleviating viable bioburden compared to intermittent DC therapy or toothpaste treatment, whilst the interkingdom biofilm model remained largely unaffected by brushing only.

## 5. Materials and Methods

All biofilm studies reported in this work were carried out in accordance with the minimum information guidelines specified for biofilm formation in microplates [35].

### 5.1. Generation of the Denture Biofilm Model

The biofilm models used and repurposed as an inoculum have previously been published by the research group [20,36,37]. Using recent microbiome data and well documented consortia of oral microorganisms, these models were created to encompass the main etiological agents of the oral diseases. Sequential addition of the organisms was used to mimic the natural progression of disease states with initial colonisers attaching to the surface, intermediate colonisers to allow the bridging between organisms and finally late pathogenic colonisers [38].

#### 5.1.1. Growth and Standardisation of Bacteria

All pure culture organisms were stored prior to use in Microbank™ beads (Pro-lab Diagnostics, UK) at −80 °C. Prior to culture preparation, organisms were revived from frozen stocks as follows: *Streptococcus mitis* NCTC 12261, *Streptococcus intermedius* ATCC 27335, *Streptococcus oralis* ATCC 35037, *Streptococcus mutans* ATCC 25175 and *Aggregatibacter actinomycetemcomitans* DSMZ 1123 were grown and maintained at 37 °C, 5% CO_2_ on Colombia blood agar (CBA, Oxoid, Basingstoke, UK). *C. albicans* 10231 was maintained on Sabouraud’s dextrose agar (Oxoid, UK) at 30 °C in aerobic conditions for 48 h. *Fusobacterium nucleatum* ATCC 1096, *F. nucleatum* ssp. *vincentii* ATCC 49256, *Actinomyces naeslundii* ATCC 19039, *Veillonella dispar* ATCC 27335, *Prevotella intermedia* ATCC 25611 and *Porphyromonas gingivalis* W83 were maintained at 37 °C on fastidious anaerobic agar (FAA, Lab M, Heywood, UK) in an anaerobic chamber (Don Whitley Scientific Limited, Bingley, UK) with an atmosphere of 85% N_2_, 10% CO_2_ and 5% H_2_ for 48 h or 72 h for strict anaerobes. All media and agar used in the anaerobic chamber was deoxygenated for 24 h prior to use. *Lactobacillus casei* DSMZ 20011 was grown and maintained on MRS agar (Oxoid, Basingstoke, UK) in 37 °C, 5% CO_2_. Aerobic organisms were then stored at 4 °C for a maximum of 2 weeks prior to propagation, while anaerobic organisms were stored within the anaerobic chamber for 1 week prior to being re-streaked.

Culture broths for propagation of *S. mitis, S. intermedius, S. oralis, S. mutans* and *A. actinomycetemcomitans* were grown overnight (16–18 h). Broths were grown statically in 10 mL tryptic soy broth (TSB, Sigma-Aldrich, Gillingham, UK) at 37 °C, 5% CO_2_. *C. albicans* was grown for 16–18 h in 10 mL yeast peptone dextrose (YPD, Sigma-Aldrich) at 30 °C in an orbital benchtop shaker at 200 rpm, 20 mM orbital diameter (IKA KS 4000 I Control, Staufen, Germany). *P. gingivalis, F. nucleatum and F. nucleatum* ssp. *vincentii* were propagated in 10 mL of deoxygenated Schaedlers anaerobic broth (Oxoid, UK). *V. dispar, A. naeslundii, and P. intermedia* were grown in 10 mL of brain heart infusion broth, (BHI, Sigma-Aldrich) in an anaerobic chamber at 37 °C, 85% N_2_, 10% CO_2_ and 5% H_2_. *L. casei* was grown in 10 mL of MRS broth (Oxoid, UK), 37 °C, 5% CO_2_. Anaerobic cultures were grown for 48 h as necessary, then pelleted by centrifugation (VWR Megastar 1.6R, US, 3000× *g* capacity). Pellets were then washed twice via resuspension in phosphate buffered saline (PBS, Sigma-Aldrich, UK). Washed cells were standardised to 1 × 10^8^ cells/mL using a spectrophotometer for bacteria, and via haemocytometer counting for *Candida* (cell count × dilution factor × volume of square = CFU/mL). Previously to determine accurate absorbance readings at 550 nm that equate to 1 × 10^8^ cells/mL for the bacteria, the Miles and Misra colony counting technique was employed by serially diluting pure cultures to determine the correct absorbance per organism [37]. *Streptococcus* spp., *V. dispar* and *L. casei* were read at an optical density (OD) of 0.5. *A. naeslundii*, *F. nucleatum*, *F. nucleatum* ssp. *vincentii*, *P. gingivalis*, *P. intermedia* and *A. actinomycetemcomitans* were read at an OD of 0.2.

#### 5.1.2. Development of Multi-Species Biofilms

The biofilm model used in this study was as previously described with small amendments to the protocol [20]. Bacteria and fungi were standardised to 1 × 10^7^ CFU/mL in an equal volume of Roswell Park Memorial Institute-1640 (RPMI) with Todd Hewitt Broth (THB) supplemented with 0.01 mg/mL hemin and 2 µg/mL menadione [38]. Inoculum biofilms were grown in a volume of 15 mL RPMI-THB in T75 cell culture flasks (ThermoFisher, Renfrew, UK), which increases the surface area more than 200-fold. Addition of standardised organisms and media changes were carried out using 25 mL serological pipettes and a pipette controller (Fisherbrand, Loughborough, UK). Biofilms were harvested using cell scrapers (Greiner, Kremsmünster, Austria) in a 15 mL RPMI-AS mixture containing 20% *v*/*v* glycerol (Sigma-Aldrich, Poole, UK) as cryoprotectant [39]. The biofilms were collected as 1 mL aliquots in cryovial form and stored at −80 °C. Initial inoculum biofilms were used at this point before freezing to assess the impact of harvesting as an initial step and identify any compositional changes upon freezing. Once the inoculum biofilms were prepared in bulk, the lag time to produce a testing plate was reduced 8–10 days, including culture inoculation and set up, to 2–5 days. Thus, a large batch of standardised inocula could be created and reconstituted when required for testing. Standardised biofilm inocula was applied to poly(methyl) methacrylate (PMMA) discs (produced in-house–13 mm^2^) contained within 24-well microtiter plates as the substrata for biofilm growth under anaerobic conditions at 37 °C for 24 h. Media was removed after 24 h using a multichannel pipette, a 45° angle and at the same point within the plate (bottom left of the well) to limit disruption of the biofilm.

### 5.2. Biofilm Treatment

Six treatment modalities were used, with all treatments prepared in hard water (HW, Sigma-Aldrich, UK). Constituents of the HW contained a mixture of two solutions, A and B. Solution A consisted of 2.12 g magnesium hexahydrate and 3.06 g calcium chloride dihydrate dissolved in 50 mL of dH_2_O. Solution B consisted of 1.75 g sodium bicarbonate dissolved in 50 mL of dH_2_O. The solutions were combined at a ratio of 3:4 and the pH was adjusted to 7.0. The treatments were as follows: non-treated controls (Figure 3A), daily brushing submersion in HW containing the denture cleanser (DC, Poligrip^®^ 3 Minute, GSK, Weybridge, UK) (Figure 3B), daily brushing following submersion in HW with addition of a DC on day 7 only (Figure 3C), daily brushing with Colgate^®^ Total (Colgate-Palmolive, Surrey, UK) or Crest^®^ 3D White toothpaste (P&G, Ohio, USA, Figure 3D) and daily brushing followed by submersion in HW only (Figure 3E). The positive control is considered as daily brushing followed by DC on the final day 7 (B), as described elsewhere (17). Non-treated controls were maintained in HW throughout the treatments. The DC pack instructions were followed for preparation of the tablets. Slurries of toothpaste were prepared by mixing 7 g with 21 mL HW via vortexing for 10 min, as defined in the ISO 11609:2017 standard. All PMMA discs were brushed as previously described [20,22]. After treatment, all discs were placed into 1 mL of Dey–Engley neutralizing broth (Sigma–Aldrich, UK) for 15 min. Treatments were repeated each day and discs were either re-incubated in 500 µL of THB:RPMI media or processed for colony forming counts.

### 5.3. Biofilm Viability Assessed via Colony Forming Unit Counts

Biofilms after 7 days were processed immediately for colony forming unit (CFU) counts. Biofilms were initially ultrasonically vibrated in 1 mL PBS in a sonication bath at 35 kHz for 10 min. The Miles and Misra technique was utilised to assess the number of viable and culturable cells within the biofilm [37], using either brain heart infusion (BHI) agar +10% horse blood for aerobes and anaerobes, or Sabouraud’s dextrose agar for yeast. BHI plates were either incubated at 37 °C for 48 h under aerobic or anaerobic conditions. SAB plates were incubated at 30 °C for 48 h before counting.

### 5.4. Confocal Laser Scanning Microscopy

Biofilms were stained with SYTO™-9 green-fluorescent stain and propidium iodide (PI) at concentrations of 20 µM prepared in dH_2_O. Biofilms were washed in dH_2_O, stained for 15 min in the dark then fixed with 4% paraformaldehyde for 30 min, protected from light. After a final wash with dH_2_O, the samples were left to air dry and the underside of the PMMA discs were affixed to a slide with superglue, prior to imaging on the Zeiss LSM780 confocal laser scanning electron microscope (ThermoFisher, UK). Z-stack images were obtained and edited using ImageJ.

### 5.5. Statistical Analysis and Data Presentation

Graph production, data distribution and statistical analyses were performed using GraphPad Prism (version 8; La Jolla, CA, USA) on 6 replicates from each experimental arm performed over two independent experiments. After assessing whether the data conformed to a normal distribution, Kruskal–Wallis tests with Dunn’s post-test comparison were used to compare groups to the control. Statistical significance was achieved upon *p* < 0.05.

## Figures and Tables

**Figure 1 antibiotics-11-00113-f001:**
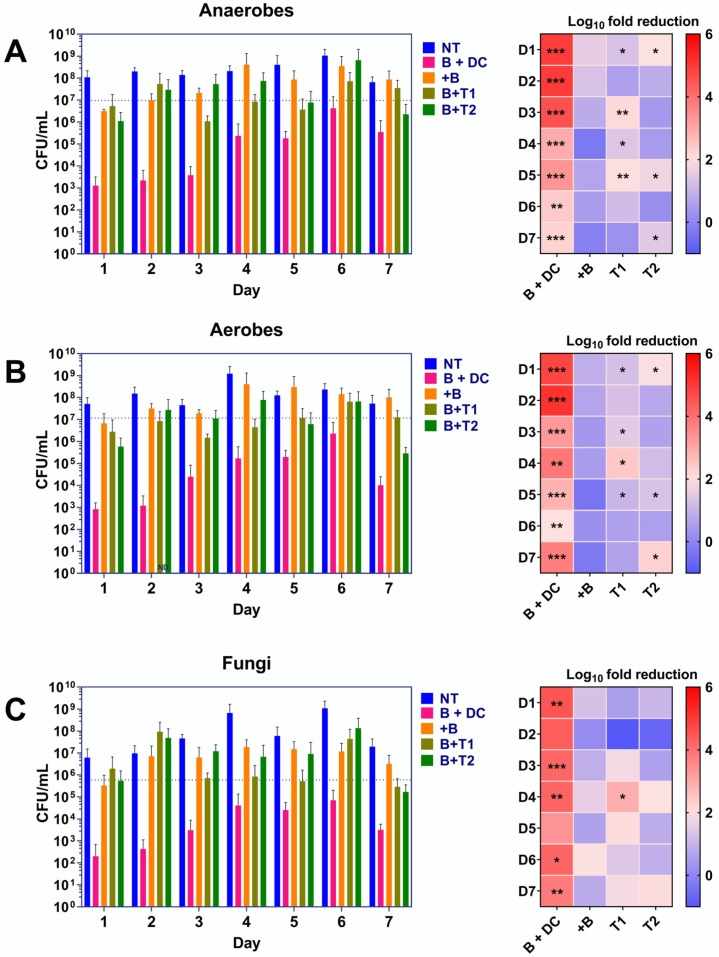
Viability counts for bacteria and fungi in the denture biofilm model. Denture biofilms formed on poly(methyl) methacrylate (PMMA) discs were treated for 7 days with six separate treatment modalities. These were non-treated controls (NT; blue bars), brushing with daily denture cleansing treatment (B + DC; pink bars), brushing only (+B; yellow bars), brushing with Colgate^®^ toothpaste (B + T1; light green bars) or brushing with Crest^®^ toothpaste (B + T2; dark green bars). The final treatment consisted of brushing with denture cleanser treatment on day 7 only (CFU counts for these are denoted by the dotted lines on the bar graphs). Results shown are representative of 2 individual experiments repeated with 3 separate replicates (*n* = 6), each for anaerobic bacteria (**A**), aerobic bacteria (**B**) and fungi (**C**). Heatmaps show the average log_10_ fold reductions in CFU counts relative to NT. Significance values are shown as follows; * *p* < 0.05, ** *p* < 0.01 and *** *p* < 0.001, all calculated relative to the NT using the Kruskal–Wallis test with Dunn’s post-test.

**Figure 2 antibiotics-11-00113-f002:**
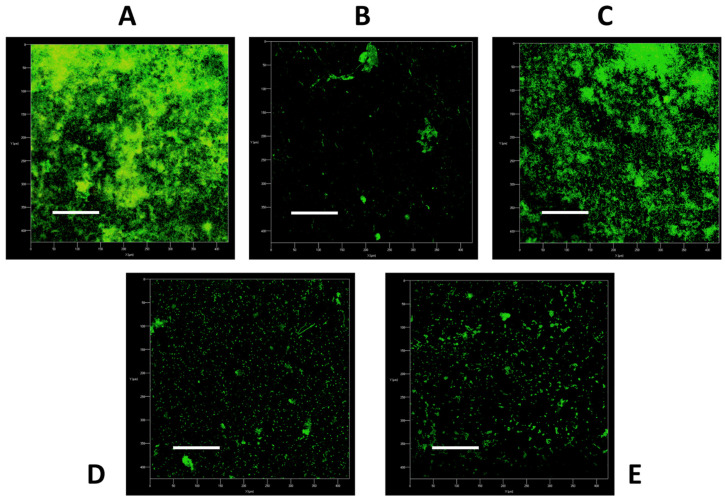
Confocal images of the non-treated and treated biofilms. Treatment regimens: (**A**), non-treated; (**B**), brushing with denture cleanser; (**C**), brushing only; (**D**), brushing with Colgate^®^; (**E**), brushing with Crest^®^. Overlaid images were representative of compiled z-stacks (a series of confocal images taken at different focal depths across the same x- and y-axes) taken from one experiment on the Zeiss LSM780 confocal laser scanning electron microscope. Scale bars represent 100 µm.

**Figure 3 antibiotics-11-00113-f003:**
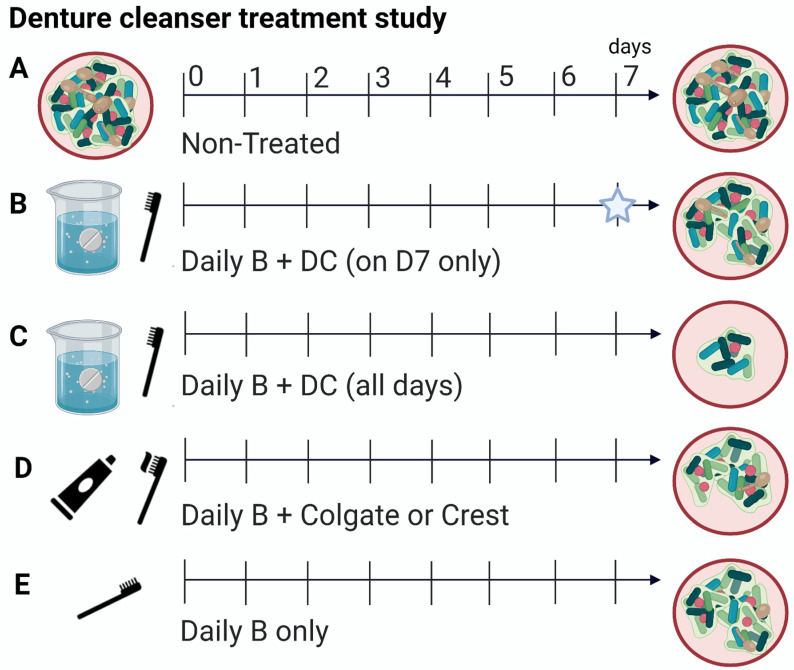
Schematic diagram depicting the treatment modalities used in this study. Polymicrobial biofilms were treated for 7 days with a variety of modalities. Biofilms were either left non-treated, (**A**) treated with daily brushing and denture cleanser (DC) on day 7 only (star represents single treatment), (**B**) daily brushing and DC on all days, (**C**) daily brushing with Colgate^®^ toothpaste or Crest^®^ toothpaste, (**D**) and finally, brushing only minus additional treatment, (**E**). Figure was created using Biorender.com (accessed on 1 November 2021).

## Data Availability

The data presented in this study are available within this article.

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
