# Peer review of "An In Vitro Evaluation of Denture Cleansing Regimens against a Polymicrobial Denture Biofilm Model"

_antibiotics, 2022, doi:10.3390/antibiotics11010113_

Round 1
Reviewer 1 Report
The manuscript by Brown et al evaluates different denture cleansing regimens against a multispecies biofilm. The manuscript is scientifically solid and well written. My major comment is that there were only two independent repeats of the main experiment (Figure 1).
Minor comments:
Lines 111 – It would be helpful to mention somewhere the type of stain used, and not only on the M&Ms.
Figure 3 – Include scale bar.
Line 238 – Why are the ODs for different microorganism read at different wavelengths?
Line 250 – 20% in vol/vol?
Line 266 – Supliers of HW components are missing.
Lines 271 – There are numerous versions of Poligrip, Colgate and Crest toothpaste. The identification of the components that were used should be clear.
Line 277 – What was the rational behind the selection of 7 g of toothpaste and 21 mL HW?
Line 280 – Which media were the discs re-incubated in?
Lines 284 – Was there any control related to the amount of biofilm that was left on the discs after sonication?
Author Response
We thank the reviewer for their helpful and insightful comments. We have edited the manuscripts, and all changes relevant to reviewer 1 are highlighted in yellow. We have also provided a response below for specific queries.
The manuscript by Brown et al evaluates different denture cleansing regimens against a multispecies biofilm. The manuscript is scientifically solid and well written. My major comment is that there were only two independent repeats of the main experiment (Figure 1).
This is a valid point. The reason for only two independent repeats for this study is down to the size, complexity and number of parameters evaluated in the experimental design, as displayed in Figure 3. Although this was performed on two occasions, 6 independent discs were sampled for each parameter, that enables a robust statistical analysis. We have made sure this is clear in the methods “Graph production, data distribution and statistical analysis were performed using GraphPad Prism (version 8; La Jolla, CA, USA) on 6 replicates from each experimental arm performed over two independent experiments.”
Lines 111 – It would be helpful to mention somewhere the type of stain used, and not only on the M&Ms.
SYTO™-9 green-fluorescent stain and propidium iodide (PI) were used to determine level of viable and dead cells (“Minimal red propidium iodide (PI) staining was detected, suggestive that all biofilms remaining on the discs were largely viable”. We have added further detail of these in section 2.2 to make it clearer what the colours and stains refer to.
Figure 3 – Include scale bar.
Apologies, this should be actually Fig 2 and has been amended accordingly. Scale bars are not required as the x and y axis already shows the dimensions of the image, which can be observed if figure enlarged as these are high resolution.
Line 238 – Why are the ODs for different microorganism read at different wavelengths? Organisms differ in morphology and size, so different OD’s are required to reach a Miles and Misra count of 1x108 allowing us to accurately standardise the organisms added.
Line 250 – 20% in vol/vol?
Yes v/v was used and has been edited for glycerol.
Line 266 – Suppliers of HW components are missing.
This has been amended to Sigma-Aldrich, UK.
Lines 271 – There are numerous versions of Poligrip, Colgate and Crest toothpaste. The identification of the components that were used should be clear.
Apologies for this obvious omission. These specific details have been added “(DC, Poligrip® 3 Minute) (Fig 3B), daily brushing following submersion in HW with addition of a DC on day 7 only (Fig 3C), daily brushing with Colgate® Total or Crest® 3D White toothpaste (Fig 3D)”
Line 277 – What was the rational behind the selection of 7 g of toothpaste and 21 mL HW?
This is a commonly used slurry mixture of that equates to 25% w/v that is derived from the industry as an approximate standard for toothpaste use and is related to the ISO 11609:2017 standard. We have added this detail in the methods.
Line 280 – Which media were the discs re-incubated in?
THB:RPMI – this detail has been added.
Lines 284 – Was there any control related to the amount of biofilm that was left on the discs after sonication?
We have discussed this issue above, and provided a sentence on limitations based on the porous nature of the material. We cannot discount this retained material, however, the series of experiments are completed in parallel and are quantitatively compared. Residual adherent cells will remain, but given the large quantities of cells then the impact on the reported cells numbers is minimal based on the overall effectiveness of sonication.
Reviewer 2 Report
Most of the comments on the attached file relate to the clarity and consistency within the manuscript. Overall- good writing and presentation.
The authors need to use more "lean" writing to have created impact. See comments. Also, many of these sentences could be inverted. That is, put the subject-verb clause first, as shown, then the dependent clause. Remember- you have international readers.
Author Response
We thank the reviewer for their extensive editing. We have carefully copy edited the suggested changes to significantly improve the manuscript. The relevant questions were completed during the edits, though specifically the queries relating to number of repeats have been addressed (3 repeats in triplicate), and the star on Figure 3 is simply to demonstrate the single end point experiment. This has been annotated on the figure legend.
Reviewer 3 Report
The authors analysed an in vitro evaluation of denture cleansing regimens against a polymicrobial denture biofilm model. The publication is not acceptable in this form. It needs major revision.
1. This special issue belongs to the Antibiotics journal. However, the authors do not mention anything about treatment of dental stomatitis using antibiotics or antifungals. Please, add a paragraph about this aspect.
2. There is also no information about positive controls. Usually positive controls are done with antibiotics or antifungals.
3. Figure 3 is illegible. What does the asterisk stand for? Please correct the diagram.
4. Besides, there are minor stylistic errors (+ form of writing without italics), etc. I recommend rereading the entire paper.
Author Response
We thank you for these valuable comments and have amended the manuscript according to your comments and the other reviewers. We have highlighted in light blue your suggested changes where possible, unless previous reviewer commented a similar point.
This special issue belongs to the Antibiotics journal. However, the authors do not mention anything about treatment of dental stomatitis using antibiotics or antifungals. Please, add a paragraph about this aspect.
We appreciate that this special issue is on antibiotics, but the term antibiotic does not need to specifically relate to a conventional approved antimicrobial. However, for balance we have included these sentences to highlight the different treatment modalities for individuals with overt disease symptoms. “Indeed, conventionally DS is treated with fluconazole, itraconazole, or nystatin (amphotericin B). These antifungal drugs that have a limited scope against Candida biofilms, and no activity against the dominant bacterial species within these communities [7,8]. Moreover, there is evidence that interkingdom interactions play a critical role in protecting fungal and bacterial species from antibiotics and antifungals [9,10].”
- There is also no information about positive controls. Usually positive controls are done with antibiotics or antifungals.
No positive controls used – however the primary aim was to compare the DC efficacy with the toothpastes, not compare with antifungals/biotics which are rarely used. Also, given that the use of antibiotics and antifungals in certain patients would be done in vivo (and not on the removed denture material), we do not see how this would fit the rationale of the study design. In real terms the brushing with toothpaste acts at the normal positive control.
- Figure 3 is illegible. What does the asterisk stand for? Please correct the diagram.
I am unsure why this is the case as the document submitted has a high resolution image. This has been reuploaded. The asterix represents the only treatment modality with a single measurement and this is now annotated in the legend.
- Besides, there are minor stylistic errors (+ form of writing without italics), etc. I recommend rereading the entire paper.
This has been done and based on an extensive copy edit by reviewer 2 then this has been improved.
Round 2
Reviewer 1 Report
The comments were well addressed and I feel that the manuscript is ready for publication. My only not of advice would be for the authors to revise Figure 2. The axis labels are very hard to read and there is no scale bar available.
Author Response
We have enhanced the quality of the figure. I believe one of the issues is when the manuscript is converted to pdf for review that the quality is compromised. The final high resolution figs are available to publisher. We have added 100um scale bars and highlighted this in the legend to ensure that this is clear.
Reviewer 3 Report
The authors answered most of my questions except the ones about positive control. You wrote that normal toothbrushing is already a positive control in itself. So please add this information in the manuscript.
I have attached below the paper in which positive control is included. After these amandments, I will approve the manuscript for publication.
Ramage G, O'Donnell L, Sherry L, Culshaw S, Bagg J, Czesnikiewicz-Guzik M, Brown C, McKenzie D, Cross L, MacInnes A, Bradshaw D, Varghese R, Gomez Pereira P, Jose A, Sanyal S, Robertson D. Impact of frequency of denture cleaning on microbial and clinical parameters - a bench to chairside approach. J Oral Microbiol. 2018 Oct 29;11(1):1538437. doi: 10.1080/20002297.2018.1538437. PMID: 30598732; PMCID: PMC6225516.
Author Response
Thanks for the positive responses. We have clarified the positive control and linked this with the earlier paper. To clarify, we should have stipulated that the standard positive control as it stands is daily brushing followed by a day 7 denture cleanser. We have added the text and reference, and highlighted this in the revised manuscript. "The positive control is considered as daily brushing followed by DC on the final day 7 (B), as described elsewhere (17)."